# A Novel Peptide Derived from the Transmembrane Domain of Romo1 Is a Promising Candidate for Sepsis Treatment and Multidrug-Resistant Bacteria

**DOI:** 10.3390/ijms22158243

**Published:** 2021-07-31

**Authors:** Deok-Gyun You, Hye-Ra Lee, Hong-Kyu Kim, Gi-Young Lee, Young-Do Yoo

**Affiliations:** 1Laboratory of Molecular Cell Biology, Graduate School of Medicines, Korea University College of Medicine, Korea University, Seoul 02841, Korea; ydg522@korea.ac.kr (D.-G.Y.); hyeraleee@gmail.com (H.-R.L.); gylative@gmail.com (G.-Y.L.); 2Department of Surgery, Seoul National University College of Medicine, Seoul 03080, Korea; limittango@gmail.com

**Keywords:** AMPR-11, Romo1, peptide antibiotics, drug resistance, multidrug-resistant bacteria, sepsis

## Abstract

The emergence of multidrug-resistant (MDR) bacteria through the abuse and long-term use of antibiotics is a serious health problem worldwide. Therefore, novel antimicrobial agents that can cure an infection from MDR bacteria, especially gram-negative bacteria, are urgently needed. Antimicrobial peptides, part of the innate immunity system, have been studied to find bactericidal agents potent against MDR bacteria. However, they have many problems, such as restrained systemic activity and cytotoxicity. In a previous study, we suggested that the K58–R78 domain of Romo1, a mitochondrial protein encoded by the nucleus, was a promising treatment candidate for sepsis caused by MDR bacteria. Here, we performed sequence optimization to enhance the antimicrobial activity of this peptide and named it as AMPR-22 (antimicrobial peptide derived from Romo1). It showed broad-spectrum antimicrobial activity against 17 sepsis-causing bacteria, including MDR strains, by inducing membrane permeabilization. Moreover, treatment with AMPR-22 enabled a remarkable survival rate in mice injected with MDR bacteria in a murine model of sepsis. Based on these results, we suggest that AMPR-22 could be prescribed as a first-line therapy (prior to bacterial identification) for patients diagnosed with sepsis.

## 1. Introduction

Antibiotics are considered one of the greatest advances of the 20th century because of the dramatic expansion of the average human lifespan they enabled [1]. However, bacteria resistant to specific antibiotics are increasing because of the abuse and long-term use of antibiotics. Particularly, the emergence of multidrug-resistant (MDR) bacteria is a health problem worldwide. Although a global action plan to solve antimicrobial resistance was published by the World Health Organization in 2015, it is still predicted that more than 10 million people will die of MDR bacterial infections in 2050 [2].

Sepsis is a critical disease with a high mortality rate [3] in which a bacterial infection triggers a potent inflammatory response throughout the body, causing tissue and organ failure. As it progresses, sepsis produces septic shock, which is an abnormal condition of cellular metabolism caused by extremely low blood pressure, which cannot replace bodily fluids [4]. Sepsis caused by an MDR bacterial infection, especially the “ESKAPE” pathogens (*Enterococcus faecium*, *Staphylococcus aureus*, *Klebsiella pneumoniae*, *Acinetobacter baumannii*, *Pseudomonas aeruginosa*, and *Enterobacter* species), is a particularly serious problem [5]. Patients with sepsis caused by MDR-bacterial strains, such as carbapenem-resistant *P. aeruginosa* (CRPA) and methicillin-resistant *S. aureus* (MRSA), have poor clinical outcomes [6]. Antibiotic therapy is the first line of sepsis treatment, and the proper medicine must be administered to patients as quickly as possible [7]. However, it is difficult to quickly identify the causal bacterium in blood [8].

Antimicrobial peptides (AMPs) are natural bactericidal agents produced by many organisms for host defense. Because they have potent antimicrobial properties with a fast-acting mechanism against various microorganisms, AMPs have been under the spotlight as a potential new class of antibiotics that could replace conventional antibiotic agents in the treatment of MDR bacterial infections [9]. Thousands of AMPs have been reported in various species, and short cationic peptides (fewer than 100 amino acids) with amphiphilic properties play roles in many of them [10]. These cationic AMPs have been suggested to bind to bacterial membranes through an electrostatic interaction, and the action mechanisms have been explained using the toroidal pore model, the barrel-stave model, and the carpet model [11]. Because of those action mechanisms, AMPs produce less bacterial resistance than conventional antibiotics, and they have a broad spectrum of activity in gram positive, gram negative, and MDR bacterial strains [12]. Although many AMPs are under investigation for clinical development, only a few have been approved by the United States Food and Drug Administration (USFDA) for clinical use [13]. One obstacle to developing AMPs for clinical use is poor correlation between in vitro and in vivo results. Therefore, it is essential to verify antimicrobial activity in an animal model when developing AMPs as drug candidates. Furthermore, it has been demonstrated that AMPs are unstable and have a short half-life in living organisms [14], and they have problems such as toxicity and immunogenicity [15]. To navigate those hurdles, various optimized or modified AMPs have been designed from natural AMPs to use their mechanism for killing bacteria [16].

Reactive oxygen species modulator 1 (Romo1) is a mitochondrial protein encoded by the nucleus, and it has been reported to have various cellular functions [17,18,19,20]. Recently, it was identified as a nonselective cation channel in the mitochondrial inner membrane [21]. It consists of two transmembrane domains (TMDs). The second TMD of Romo1 forms pores with an amphipathic helical structure in biological membranes [21]. In a previous study, we reported that this second TMD, named AMPR-11 (antimicrobial peptide derived from Romo1), has a broad spectrum of antimicrobial activity, including against MDR bacteria [22]. Even though the efficacy of AMPR-11 in a murine sepsis model was satisfactory, the in vitro and in vivo antimicrobial activity of AMPR-11 needed to be improved to make it suitable for potential development as a new antibiotic. In this study, we used substitution and deletion to optimize the peptide sequence of AMPR-11, and we named the new form AMPR-22 (antimicrobial peptide derived from Romo1). This optimized peptide from AMPR-11 not only exhibited remarkably enhanced antimicrobial activity in vitro and in vivo but also showed decreased hemolytic activity in human blood. Given the results of this study, we suggest that AMPR-22 is a promising therapeutic option for MDR bacterial infections and complications.

## 2. Results

### 2.1. Improving the Antibacterial Activity of AMPR-11 through Sequence Substitution and Deletion

In a previous study, we reported that AMPR-11 had an amphipathic structure typical of AMP [22]. It contains uncharged amino acid residues such as Thr^2^, Gln^5^, Ser^6^, and Thr^9^ on one side of its helix surface. Because AMPR-11 is not highly soluble in physiological buffer conditions, we sequentially substituted those uncharged amino acids with Lys. We expected those changes to also improve the antibacterial activity of AMPR-11 because positively charged amino acids such as Arg and Lys play an important role in most AMP activity [23,24]. As we expected, sequentially substituted analog peptides with Lys (K2, K3, and K4) showed gradually improved antibacterial properties in a bacterial killing test by the modified minimum bactericidal concentration (MBC) determination method (Table 1). We also sequentially deleted one amino acid from Lys^1^ to Arg^21^ based on the K4 peptide, which showed the best MBC value among the substitution analog peptides. All the deletion analog peptides except d70F and d73I exhibited enhanced antibacterial activity. Next, we preliminarily tested the in vivo activity of the deletion analog peptides, and we found that the d75M peptide, in which Met^18^ was deleted, showed the best antibacterial activity in a murine model of sepsis (data not shown). The K4 peptide and d75M peptide were then named AMPR-21 and AMPR-22, respectively.

### 2.2. Physical Characterization and Secondary Structure Analysis of the AMPR-11 Analog Peptides

The physical properties of AMPR-21 and AMPR-22 were examined based on their amino acid sequences, as shown in Figure 1A. These peptides exhibited an increased positive charge, more hydrophilicity, and an increased hydrophobic moment compared with AMPR-11. In the helical wheel projections, the polar amino acids of these peptides are concentrated on one side surface (Figure 1B). To predict the secondary structure of AMPR-21 and AMPR-22, we performed a circular dichroism (CD) analysis in various buffer conditions. First, we used 50% hexafluoro-2-propanol (HFIP), which is commonly used in determining a protein’s secondary structure [25], and those results demonstrated that the representative structure of these peptides is an alpha helix (Figure 1C,D).

### 2.3. AMPR-22 Showed Broad-Spectrum Antibacterial Activity by Disrupting Bacterial Membrane Integrity

AMPR-11 showed antibacterial activity in various bacterial strains, including MDR strains [22]. AMPR-21 and AMPR-22 were also expected to show antibacterial activity. Therefore, we determined the MBC value of these peptides against the 17 bacterial strains we tested previously [22] by modified MBC determination assay. As expected, both peptides showed a broad spectrum of antibacterial activity (Table 2). Next, we measured cell permeabilization caused by AMPR-11, AMPR-21, and AMPR-22 using a propidium iodide (PI) uptake assay and flow cytometry. AMPR-22 induced more cell permeabilization than AMPR-11 or AMPR-21 in both gram-positive and gram-negative bacterial strains (Figure 2A,B). The outer membrane permeabilization in gram-negative bacteria was also analyzed using a 1-N-phenylnaphthylamine (NPN) uptake assay. NPN is normally not permeabilized by the bacterial membrane and is weakly fluorescent in a buffer solution, but when it is taken up by membrane permeabilization, the fluorescence intensity increases strongly. Therefore, the fluorescence intensity of NPN is detectable only in the periplasm of bacteria [26]. As shown in Figure 2C, AMPR-22 treatment immediately induced NPN uptake in a concentration- and time-dependent manner. Next, we measured AMPR-22-induced membrane depolarization in a species of gram-positive bacteria, *S. aureus*, using 3,3′-dipropylthiadicarbocyanine iodide, {DiSC_3_(5)}, a membrane potential-sensitive dye [27]. In this assay, AMPR-22 treatment induced a rapid fluorescence increase in a concentration-dependent manner (Figure 2D), indicating that AMPR-22 depolarized the bacterial membrane potential. The leakage of adenosine triphosphate (ATP), which is the main energy source inside cells, was also examined after AMPR-22 treatment of gram-positive and -negative bacteria. Extracellular ATP levels increased after AMPR-22 treatment in both bacterial strains (Figure 2E). Those results demonstrate that AMPR-22 treatment triggers disruption of both the outer and inner bacterial membranes. This bacterial disruption was directly observed in CRPA and MRSA using electron microscopy. In a scanning electron microscope (SEM), the membranes of the MDR bacterial strains showed obvious morphological changes. The membrane surface of bacteria treated with AMPR-22 was severely wrinkled and shrunk (Figure 2F). Morphological changes were also observed using transmission electron microscopy (TEM). In that experiment, the bacterial membrane was damaged and disrupted by AMPR-22 treatment (Figure 2G). In addition to those experiments, we performed a serial passage assay to determine whether AMPR-22 would enable the acquisition of drug resistance. As shown in Figure 2H, *P. aeruginosa* showed high sensitivity to AMPR-22 for 30 passages. In contrast, resistance to gentamicin developed after nine passages. The modified MBC value of bacteria treated with gentamicin increased 64-fold after 30 passages (Figure 2H).

### 2.4. The Cytotoxicity and Hemolytic Activity of AMPR-22

We compared the toxicity of AMPR-22 in mammalian cells with that of AMPR-11 and AMPR-21. AMPR-22 showed slight cytotoxicity against HeLa and human embryotic kidney (HEK) 293 cells, similar to the results with AMPR-11 (Figure 3A). However, AMPR-22 showed less toxicity than magainin 2, a well-known AMP [28], and daptomycin, a lipopeptide antibiotic [29]. In a study of hemolytic activity against mouse erythrocytes, AMPR-22 exhibited relatively endurable hemolysis properties at concentrations up to 256 µg/mL (Figure 3B). Interestingly, AMPR-22 and AMPR-21 showed negligible hemolytic activity against human erythrocytes up to 256 µg/mL, although AMPR-11 showed low hemolytic activity at 256 µg/mL. To investigate the toxicity of AMPR-22 to white blood cells, we isolated fresh peripheral blood mononuclear cells (PBMCs) from humans and mice. AMPR-22 showed negligible cytotoxic activity against human and mouse leukocytes in a flow cytometry analysis (Figure 3C). To check for in vivo toxicity before administering AMPR-22 to mice in a murine model of sepsis, we intravenously injected 100 mg/kg single dose of AMPR-22 (10-fold higher than the working dose) into BALB/c mice and tracked them for 15 days. No significant changes, such as weight loss or death, developed compared with PBS-treated mice (Figure 3D).

### 2.5. Assessment of the Antibacterial Activity of AMPR-22 in Various Conditions

AMPR-11 has fast-acting antibacterial activity that can be affected by various factors [22]. In this study, we investigated the antibacterial activity of AMPR-22 in various conditions. In mouse plasma and human serum, the antibacterial properties against both bacterial strains (*P. aeruginosa* and *S. aureus*) decreased over time, but the half-life of antibacterial activity for AMPR-22 was longer than that of AMPR-11 (Figure 4A,B). In addition, the antibacterial activity of AMPR-22 was not affected by human low-density lipoprotein (LDL), high-density lipoprotein (HDL), or bovine serum albumin, which is consistent with the results from AMPR-11 (Figure 4C,D). To estimate the antibacterial activity of AMPR-22 in whole blood, AMPR-22 was incubated with whole human and mouse blood for 180 min, and the antibacterial activity was measured using a colony-counting assay at each time point (Figure 4E). Next, we examined the in vivo efficacy of AMPR-22 in mice intravenously injected with bacteria. One hour after the bacterial injection, AMPR-22 was injected into the mice, and whole blood was collected from the tail vein at various time points (15, 30, 60, and 90 min) for colony-counting assays (Figure 4F,G). After 90 min, the number of bacteria increased by more than 400% in the PBS-treated control group, whereas it decreased by up to 50% in mice treated with AMPR-22. To estimate the functional half-life of AMPR-22, we collected blood at various time points from the tail veins of mice injected with AMPR-22 and used it for colony-counting assays. As shown in Figure 4H, more than 50% of the antibacterial activity of AMPR-22 was maintained 37 min after peptide injection. Therefore, we suggest that AMPR-22 has good antibacterial activity in vivo.

### 2.6. Efficacy of AMPR-22 in a Murine Model of Sepsis Caused by MDR Bacteria

We compared the efficacy of AMPR-22 with that of AMPR-11 and AMPR-21 in a murine model of sepsis generated by intravenously injecting mice with *S. aureus*. All peptides were administered as a single dose of 10 mg/kg one hour after bacterial injection. Interestingly, the survival rate of mice that received AMPR-22 was more than 90% (Figure 5A). The efficacy of AMPR-22 was also observed in a model of *P. aeruginosa* infection, where it had a 100% survival rate (Figure 5B). To examine murine sepsis caused by MDR bacteria, we performed the same experiments with both MRSA and CRPA and again found a mouse survival rate of 100% (Figure 5C,D). These results indicate that the efficacy of AMPR-22 in a murine model of sepsis is better than that of AMPR-11 and AMPR-21. Single-dose administration worked successfully in a murine model of sepsis caused by various bacterial strains, including MDR bacteria.

Next, we verified the antibacterial activity of AMPR-22 in a murine model of sepsis with multiple injections. AMPR-22 (3 mg/kg) was intravenously injected into the septic mice three times at one-hour intervals. As shown in Figure 5E, the survival rate with multiple doses was better than that with a single dose. It is well known that septic shock is a common cause of death in sepsis and that increased levels of cytokines are frequently observed in patients [30]. To investigate the effect of AMPR-22 on mouse cytokine levels, we performed an enzyme-linked immunosorbent assay (ELISA) 18 h after CRPA infection. AMPR-22 significantly reduced the levels of pro-inflammatory cytokines (IL-6 and TNF-α) and an anti-inflammatory cytokine (IL-10) (Figure 5F).

## 3. Discussion

The emergence of MDR bacteria through the abuse and long-term use of antibiotics is a serious health problem worldwide. To solve this problem, many investigations have been conducted, and one possibility is to develop AMPs, which are also known as host defense proteins, because of their antimicrobial properties. However, AMPs have some issues related to in vivo stability, toxicity, and manufacturing cost. Therefore, few clinical trials have had positive outcomes [31,32]. Many approaches have been investigated to improve their efficacy, including sequence optimization from natural AMPs to produce synthetic AMPs [33]. For example, magainin is a widely studied AMP with a typical amphipathic structure. It has hydrophobic residues on one side of its helical structure and polar residues on the other [34]. Many studies to improve its antimicrobial activity have investigated its structural features [35,36]. LL-37 is also a well-studied human AMP that has been studied extensively [37]. Although many investigations have been conducted, few peptides are available for clinical use. Daptomycin, vancomycin, and telavancin are AMPs that have been approved by the FDA. However, they lack a broad spectrum of antimicrobial activity [32], so they are not suitable for sepsis treatment. Sepsis is a systemic inflammation that can be triggered by many pathogens, including gram-positive bacteria, gram-negative bacteria, and fungi [22]. In our previous study, we reported that a Romo1-derived peptide (K58–R78 region of Romo1, AMPR-11) exhibited a broad spectrum of antibacterial activity with low toxicity [22], including in a murine model of sepsis. Interestingly, Romo1 is not a host defense protein, but a mitochondrial protein encoded by the nucleus [21]. It has been reported to be a nonselective cation channel, and it consists of two TMDs [21]. Its secondary TMD has a pore-forming property and seems to have the antibacterial activity. In a murine model of sepsis, however, the survival rate of mice treated with a single dose (10 mg/kg) of that secondary TMD, AMPR-11, was around 60% [22]. Our aim in this study was to optimize the sequence of AMPR-11 to improve that murine survival rate. Many AMPs have been reported to have an alpha helical structure, and it has been documented that the antimicrobial activity of AMPs was enhanced by increasing the positive charge of some cationic AMPs [38,39]. To increase of antibacterial activity of AMPR-11, we changed four uncharged hydrophilic amino acids to positively charged Lys and named the resulting peptide AMPR-21. AMPR-21 and the analog peptide from which one amino acid was deleted, AMPR-22, showed biophysical properties typical of cationic, alpha helical AMPs (Figure 1), and their antibacterial activity against various bacterial strains was improved over that of AMPR-11 (Table 2). Optimization of physical properties and the deletion of amino acids could be considered when designing other AMPs. The broad-spectrum antibacterial effect of AMPR-22 could be explained by the direct interaction between AMPR-22 and bacterial membranes. Our results in this study show that AMPR-22 disrupts the integrity of bacterial membranes (Figure 2). Our previous work identified the interaction between AMPR-11 and phospholipids on the bacterial membrane, and we suggested that it triggered membrane permeabilization by binding with lipid A or cardiolipin [22]. Nonetheless, the exact binding mechanism between AMPR-22 and bacterial membranes remains to be determined.

One of main obstacles in developing new antibiotics from AMPs is their cytotoxicity. Because peptides developed from AMPs could affect human cells, including blood cells, the administration of AMPs in clinical trials has mainly focused on topical administration to treat skin infections [40]. In this study, we performed hemolytic assays for AMPR-21 and AMPR-22, and they showed less toxicity than AMPR-11 (Figure 3). Interestingly, the toxicity of AMPR-22 to human erythrocytes was lower than that in mice, although the reason for that remains to be studied. Nonetheless, this result indicates that AMPR-22 could be administered to human blood.

The other main hurdle for drug development using AMPs is their short half-life in a host organism, which is caused by proteolytic degradation or renal clearance. In this study, we found that the functional half-life of AMPR-22 was around 90 min in vitro and around 37 min in vivo (Figure 4). Although the half-life of AMPR-22 in blood is not long compared with other peptides, a single-dose treatment of the peptide (10 mg/kg) produced an almost 100% survival rate in a murine model of sepsis. Two experimental results explain why AMPR-22 has good antibacterial activity in vivo. First, AMPR-22 exhibited fast-acting antibacterial activity. In a previous study, we showed that AMPR-11 induced membrane permeabilization that released most cytoplasmic contents within 10 min [22]. Second, the antibacterial activity of AMPR-22 in vitro was about 25 times higher than that of AMPR-11.

In our previous study, we reported that a single dose (10 mg/kg) of AMPR-11 produced a 60% survival rate in a murine model of sepsis [22]. As shown in Figure 5, AMPR-22 showed an enhanced survival rate at the same dose. We expect that a lower dose of AMPR-22 might work in clinical trials because our murine model of sepsis involved injecting from 8 × 10^7^ to 1 × 10^8^ CFU of bacteria into the tail veins of mice, which is a much higher dose than typically found in human sepsis (100 CFU/mL in blood) [22]. We also expect that combining AMPR-22 with conventional antibiotics could be another therapeutic option for treating human sepsis.

A research charity, Wellcome Trust, has warned that the emergence of MDR bacteria is a global problem and that MDR bacteria will be a leading cause of death in humans by 2050 [22]. Continuous treatment with AMPR-22 did not produce drug resistance in a serial passage assay (Figure 2H). In conclusion, we suggest that AMPR-22 is a promising therapeutic candidate for the treatment of diseases caused by various bacterial strains, such as sepsis or infection with MDR bacteria.

## 4. Materials and Methods

### 4.1. Chemicals and Peptides

Daptomycin, bovine serum albumin, gentamicin, human HDL, human LDL, human serum, mouse plasma, and all other chemicals were purchased from Sigma-Aldrich (St. Louis, MO, USA). AMPR-11 and magainin 2 were chemically synthesized using solid-phase synthesis at GL Biochem (Shanghai, China). Peptides were purified to at least 75% by high-performance liquid chromatography and confirmed by mass spectroscopy.

### 4.2. Bacterial Strains

*Acinetobacter baumannii* (ATCC 19606), *B. subtilis* (ATCC 6633), *E. aerogenes* (ATCC 13048), *E. coli* (ATCC 25922), *E. faecalis* (ATCC 19433), *E. faecium* (ATCC 19434), *K. pneumoniae* (ATCC 13883), *P. aeruginosa* (ATCC 27853), *S. aureus* (ATCC 29213), *S. sindenensis* (ATCC 23963), and MRSA (ATCC 33591) were purchased from the American Type Culture Collection (ATCC; Gaithersburg, MD, USA). *Streptococcus pneumoniae* (NCCP 14585), vancomycin-resistant *E. faecium* (NCCP 11522), and vancomycin-resistant *S. aureus* (NCCP 15872) were purchased from the National Culture Collection for Pathogens (NCCP; Cheongju, Korea). The isolation and use of carbapenem-resistant bacterial strains from a patient were approved (2015AN0129) by the Institutional Review Board (IRB) of Korea University Hospital. CRPA, CRKP, and CRAB were clinically isolated in Korea University Hospital (Seoul, Korea). CRPA was confirmed to be resistant to piperacillin, piperacillin-tazobactam, ceftazidime, imipenem, meropenem, gentamicin, amikacin, and ciprofloxacin. CRAB was confirmed to be resistant to piperacillin, piperacillin-tazobactam, cefepime, ceftazidime, imipenem, meropenem, gentamicin, amikacin, and ciprofloxacin. CRKP was confirmed to be resistant to piperacillin-tazobactam, cefepime, ceftazidime, imipenem, gentamicin, and ciprofloxacin. All strains were stored at –80 °C in 50% glycerol and 50% Luria-Bertani (LB) or tryptic soy (TS) broth, grown on LB or TS plates, and aerated at 37 °C.

### 4.3. Determination of the Minimum Bactericidal Concentration

The minimum bactericidal concentration (MBC) was determined as previously described with minor modifications [22]. Briefly, bacteria were cultured in the mid-log phase and diluted to 5 × 10^5^ colony-forming unit (CFU)/mL in 10 mM sodium phosphate buffer (pH 7.4) with a 1% volume of TS broth. They were incubated for 1 h with various peptides which were two-fold diluted serially from 64 μg/mL to 0 μg/mL concentrations in a 96-well plate. The samples were inoculated TS agar plate and incubated overnight at 37 °C. The value of MBC was determined at the lowest concentrations at which no colonies were identified.

### 4.4. Circular Dichroism

The secondary structure of the AMPR-11 analogs was measured using a Chirascan CD spectrometer (Applied Photophysics, Leatherhead, UK). The peptides were dissolved at a concentration of 100 μM in 50% HFIP solution and loaded in a 1 mm pathlength quartz cuvette. The CD spectra of the samples were recorded between 180 and 260 nm, and the data were calibrated with the background scattering from a buffer-only sample. The alpha helix wheel projections were reproduced on the basis of results from the EMBOSS: pepwheel tool [41], and the secondary structures were predicted by CD spectral data using the BeStSel server [42].

### 4.5. Bacterial Membrane Permeabilization Assays

For the PI uptake assay, bacteria were cultured in the mid-log phase and diluted to 5 × 10^5^ CFU/mL in PBS (pH 7.4). The peptides were treated with 40 μg/mL for 10 min, and then 10 μM PI was added and incubated for 20 min. The PI fluorescence was measured with a flow cytometric assay using a FACS Canto II (BD Biosciences, CA, USA), and the data were analyzed using FlowJo software (Tree Star, Ashland, OR, USA). NPN uptake and a membrane depolarization assay were performed as previously described, with some changes [43]. Briefly, 1 × 10^4^ CFU/mL of bacteria were place on a 96-well plate, and then 10 μM NPN or 4 μM DiSC_3_(5) was added, and AMPR-22 was administered. The fluorescence signal was measured for 10 min using a SpectraMax Plus 384 microplate reader (Molecular Devices, San Jose, CA, USA). For the NPN uptake assay, AMPR-22 was administered 3 min after the start of the fluorescence measurements. The ATP leakage assay was performed using a BacTiter-glo™ kit (Promega, Madison, WI, USA) and the manufacturer’s instructions with minor modifications. The bacteria were cultured and diluted to 5 × 10^4^ CFU/mL in PBS (pH 7.4), and AMPR-22 was administered for 10 min. Then, the supernatant was collected by centrifugation (10,000× *g* for 5 min) and mixed with ATP assay medium. The luminescence signal was measured using a SpectraMax Plus 384 microplate reader (Molecular Devices, San Jose, CA, USA). The serial passage assay was performed as previously described with some changes [44,45]. Simply, the MBC value was determined on a TS agar plate cultured in broth medium for 24 h. Sub-MBC concentrations of peptide were administered to 1 mL of cultured cells, and then the MBC values were determined for every passage. The antibiotic resistance of bacteria was compared with that of cells not treated with drugs and cultured in fresh medium.

### 4.6. Electron Microscopy

Bacteria were grown to 1 × 10^9^ CFU and incubated with 0.2 mM AMPR-22 for an hour in culture medium, and then they were centrifuged at 1000× *g* for 5 min. TEM samples were fixed with 2% paraformaldehyde and 2.5% glutaraldehyde in 0.1 M phosphate buffer (pH 7.4) for 18 h at 4 °C. The samples were then washed two times with the same buffer used in fixation and postfixed with 2% osmium tetroxide for 90 min. After standard dehydration in ethanol (60, 70, 90, 95, and 100%), the fixed samples were infiltrated and embedded in a propylene oxide and Epon mixture. They were then cut into 70-nm sections using an FC7 ultramicrotome (Leica, Vienna, Austria) and placed on double grids. TEM images were made using a H-7650 transmission electron microscope (Hitachi, Tokyo, Japan) with 80 kV acceleration voltage. For SEM, the bacteria were treated with AMPR-22 in the same manner as for TEM. The samples were pre-fixed with medium (2.5% glutaraldehyde in 0.1 M phosphate buffer at pH 7.4) for 18 h at 4 °C. After centrifugation at 1000× *g* for 5 min, the pellet was washed two times for 20 min with the same buffer. Samples were postfixed with 2% osmium tetroxide for 2 h and then subjected to standard dehydration in ethanol (60%, 70%, 90%, 95%, and 100%). The samples were air dried, attached to a stub, and coated with platinum using an E-1045 ion sputter (Hitachi, Tokyo, Japan). The SEM images were observed using a S-4700 scanning electron microscope (Hitachi, Tokyo, Japan).

### 4.7. Cell Viability Assay

Mammalian cell viability was measured using a 3-(4,5-dimethylthiazol-2-yl)-2,5-diphenyltetrazolium bromide (MTT) assay in HeLa and HEK 293 cells. Cells were cultured on 96-well plates and incubated separately with AMPR-11, AMPR-21, AMPR-22, magainin 2, or daptomycin. After incubation for an hour, the cells were incubated with MTT solution (2 mg/mL of MTT in PBS) for an hour, and the media were removed, and dimethyl sulfoxide was added to allow the MTT formazans to solubilize. The optical density was measured using a SpectraMax Plus 384 microplate reader (Molecular Devices, San Jose, CA, USA) at 550 nm.

### 4.8. Hemolysis and White Blood Cell Toxicity Assay

For the hemolytic assay, mouse and human whole blood was diluted with PBS (pH 7.4) and centrifuged for 10 min at 500× *g*. After removing the supernatant and rinsing the pellet three times with PBS, we resuspended the erythrocytes in PBS at 2% volume. The blood solution was added to the solution of peptides with two-fold serial dilution on a 96-well plate. After incubation for an hour at 37 °C, the plate was centrifuged at 2500× *g* for 10 min, and the absorbance of the supernatant was measured at 550 nm. The PBS-treated sample was the negative control, and the 0.1% Triton X-100–treated sample was the positive control. To measure white blood cell toxicity, mouse and human PBMCs were freshly isolated as previously described [46], and 70% ethanol (positive control) or AMPR-22 was administered using two-fold diluted concentrations (0 to 256 μg/mL) for 30 min. To measure cell death, a LIVE/DEAD^®^ fixable green dead cell stain kit (Thermo Fisher Scientific, Waltham, MS, USA) was used following the manufacturer′s instructions, and flow cytometry was performed using a FACS Canto II (BD Biosciences, CA, USA). The toxicity of AMPR-22 was compared with negative (PBS-treated) and positive controls, and the data were analyzed using FlowJo software (Tree Star, OR, USA).

### 4.9. Antibacterial Activity Assays

We used the CFU counting method to measure antibacterial activity, with all bacteria cultured to the mid-log phase in an appropriate broth medium. Bacterial CFUs were counted after overnight culture at 37 °C on TS or LB agar plates. The MBC value and antibacterial activity in blood components were determined as previously described [22]. Mouse and human whole blood were collected and diluted with PBS (pH 7.4) to measure the ex vivo antibacterial activity of AMPR-22. For that test, AMPR-22 (10 μg) was added to each whole blood solution and incubated at 37 °C with shaking at 200 rpm. At each time point, it was incubated with bacteria (1 × 10^3^ CFU) for 30 min. Each sample was inoculated on an LB agar plate, and CFU counting was performed. To measure the in vivo activity of AMPR-22, 1 × 10^8^ CFU of bacteria were injected into BALB/c mice, and an hour later, 10 mg/kg of the test peptides were injected. Blood samples were collected by tail cut, serially diluted with PBS (pH 7.4), and inoculated on LB-agar plates, followed by CFU counting. To estimate the functional half-life of AMPR-22, 40 mg/kg of each tested peptide were injected into mice. Whole blood sampling was performed by cardiac puncture, and the plasma was isolated by centrifugation (2000× *g*, 15 min). The isolated plasma was incubated with bacteria (1 × 10^3^ CFU) for an hour, and then the CFU counting method was performed.

### 4.10. Murine Model of Sepsis

Specific pathogen-free (SPF) male BALB/c mice (10 weeks old; weight, 25 to 28 g) were obtained from Orient Bio (Seongnam, Korea). All mice were housed in regulated conditions (21 ± 2 °C; 50 ± 5% humidity; 12 h/12 h of light/dark cycle with light on from 8 AM, free access to water and food). After a week-long quarantine period, the mice were transferred to an animal biosafety level 2 facility with the same conditions. Injections of bacteria or peptide were intravenous, used 1 mL, 30-gauge syringes, and no individual injection exceeded 0.1 mL. The concentration of each bacterium used in the sepsis model was 8 × 10^7^ CFU (*P. aeruginosa* and *S. aureus*) or 1 × 10^8^ CFU (MRSA and CRPA). The peptides were administered an hour after bacterial infection. Survival curves were assessed for 15 days, and the data were analyzed using GraphPad Prism 7 (GraphPad Software, San Diego, CA, USA). The animal experiments were planned, approved (KOREA-2020-0087), and conducted under the guidance of the Institutional Animal Care and Use Committee at Korea University College of Medicine.

### 4.11. Mouse Cytokine ELISA

Mice were infected with CRPA as for the murine model study, and then AMPR-22 or PBS was administered an hour after infection. The whole blood of the mice was collected by cardiac puncture 18 h after the bacteria injection, and the plasma was isolated by centrifugation for 15 min at 2000× *g*. The levels of IL-6, TNF-α, and IL-10 were measured using a mouse ELISA kit (Thermo Fisher Scientific, Waltham, MS, USA) and following the manual supplied by the manufacturer. The effects of AMPR-22 were compared with PBS-treated CRPA-infected mice and non-infected mice.

## Figures and Tables

**Figure 1 ijms-22-08243-f001:**
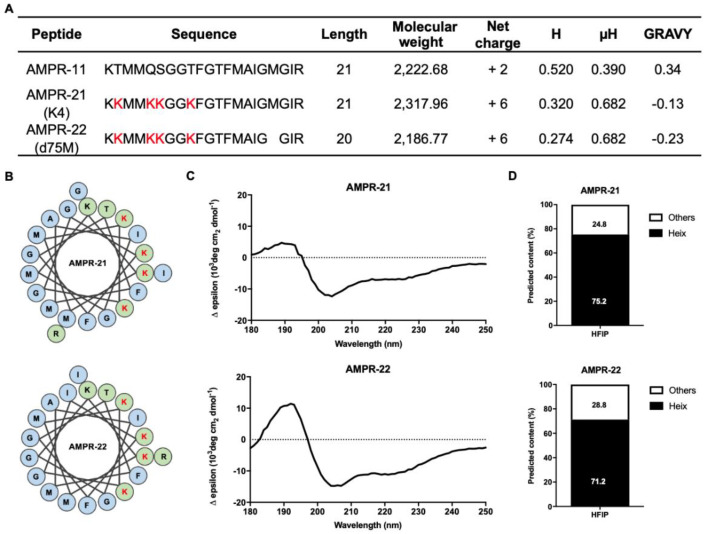
Physical properties and secondary structure analysis of the AMPR-11 analog peptides. (**A**) The physical properties of the AMPR-11 analog peptides. All items were calculated using the ProtParam tool and ADP3 server. H, hydrophobicity; μH, hydrophobic moment; GRAVY, grand average hydrophobicity and hydrophilicity. (**B**) Alpha-helical wheel prediction for AMPR-21 and AMPR-22. Illustrations were recreated based on results from the helical wheel projections server. Blue, nonpolar amino acids; green, polar amino acids; red letter, changed sequences from AMPR-11. (**C**) CD spectroscopy of AMPR-21 and AMPR-22. All of the results were corrected by the buffer-only baseline. (**D**) The secondary structure compositions of the AMPR-11 analog peptides from the CD spectra. The data were predicted using the BeStSel server.

**Figure 2 ijms-22-08243-f002:**
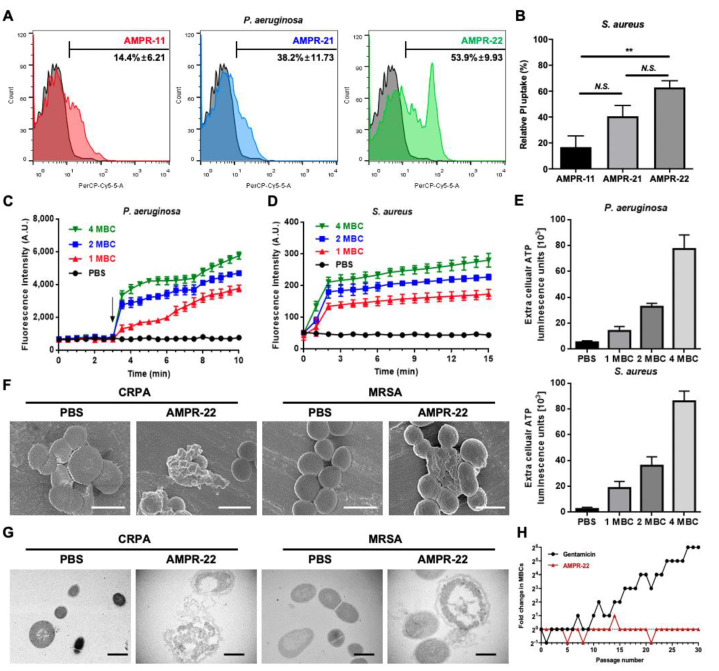
Disruption of bacterial membrane integrity by AMPR-22. (**A**,**B**) Bacterial cell death caused by peptide treatment (20 μM) was measured using a PI uptake assay with flow cytometry. (**C**) AMPR-22 induced bacterial outer membrane permeabilization. NPN uptake was measured as a fluorescence signal for 10 min after peptide treatment, which began at 3 min (arrow). (**D**) The bacterial membrane depolarization caused by AMPR-22 was measured using DiSC_3_(5) fluorescence for 15 min. (**E**) ATP leakage caused by AMPR-22 was measured using an ATP luminescence assay and a spectrophotometer. Data represent the means ± SD of more than three independent experiments. SEM images (**F**) and TEM images (**G**) of CRPA or MRSA treated with AMPR-22. Scale bar, 1 μm. (**H**) Serial passage assay of AMPR-22 and gentamicin (30 passages). ** *P* ≤ 0.01 by 2-way analysis of variance (ANOVA). *N.S.*, no significant statistical difference.

**Figure 3 ijms-22-08243-f003:**
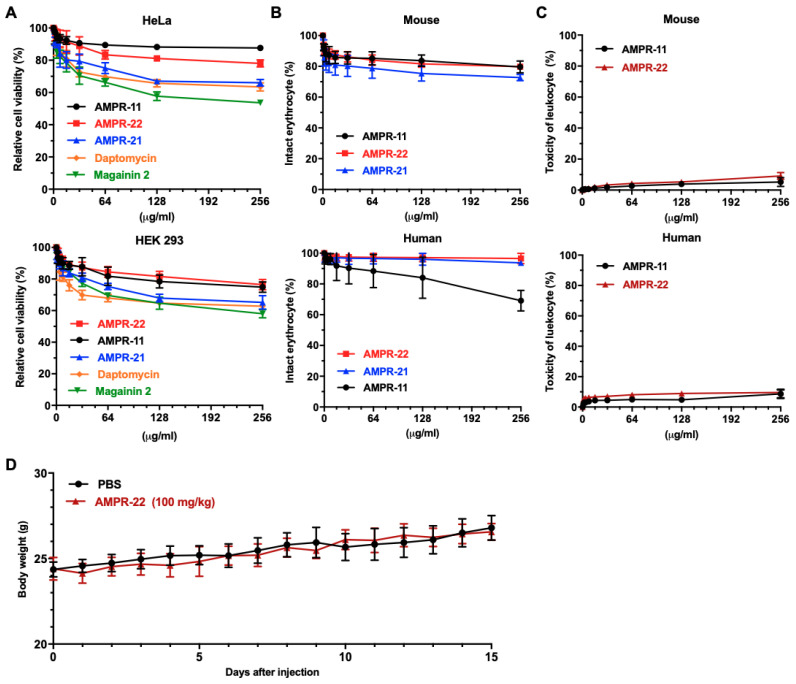
The cytotoxicity of AMPR-22. (**A**) The cell toxicity of the AMPR-11 analog peptides was measured using the MTT assay in HeLa and HEK293 cells. Daptomycin and magainin 2 were also tested. (**B**) A hemolysis assay was performed in mouse and human blood. The percentage of hemolyzed erythrocytes was measured using a spectrophotometer. Peptides were serially diluted and incubated with cells for an hour. (**C**) The toxicity of leukocytes was measured using a flow cytometric assay with LIVE/DEAD fixable dead cell™ stain dye and freshly isolated mouse and human PBMCs. (**D**) Changes in the body weight of BALB/c mice were observed for 15 days after a single intravenous administration of AMPR-22 or PBS (100 mg/kg, 10-fold higher than working concentration in survival experiments). Data represent the means ± SD of more than three independent experiments.

**Figure 4 ijms-22-08243-f004:**
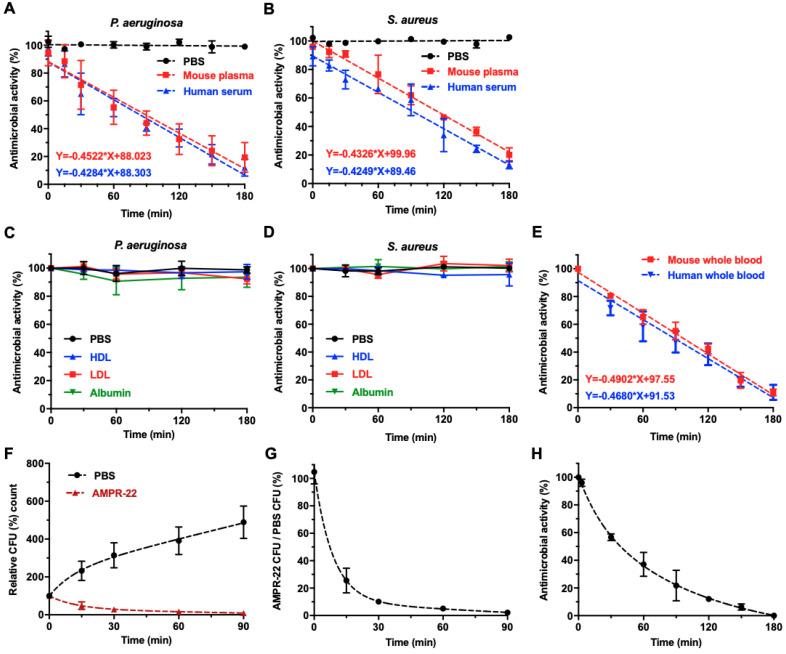
The antibacterial activity of AMPR-22 in various physiological conditions. (**A**,**B**) Effects of AMPR-22 activity in mouse plasma and human serum. AMPR-22 was incubated with blood components for the indicated times (0, 15, 30, 60, 90, 120, 150, and 180 min) and then with *P. aeruginosa* or *S. aureus* for an hour. (**C**,**D**) Effects of AMPR-22 activity in low-density lipoprotein (LDL), high-density lipoprotein (HDL), and bovine serum albumin. AMPR-22 was incubated with blood components for the indicated times (0, 30, 60, 120, and 180 min) and then with *P. aeruginosa* or *S. aureus* for an hour. (**E**) Effects of AMPR-22 activity in whole blood. AMPR-22 was incubated with mouse and human whole blood for the indicated times (0, 30, 60, 90, 120, and 180 min) and then with *P. aeruginosa* for an hour. (**F**,**G**) The antibacterial activity of AMPR-22 in vivo. AMPR-22 was intravenously administered into *P. aeruginosa*-infected mice, and blood was collected from the tail veins of the mice at the indicated times (0, 15, 30, 60, and 90 min). (**F**) The relative CFU in blood samples was calculated using the following formula: CFU at indicated time ÷ CFU at 0 min × 100 (%). (**G**) The relative CFU of blood samples obtained from F was re-graphed using the following formula: CFU of AMPR-22-treated mouse group ÷ CFU of control (PBS) group × 100 (%). (**H**) The functional half-life measurement of AMPR-22. AMPR-22 was intravenously administered into mice, and blood was collected at the indicated times (0, 5, 30, 60, 90, 120, 150, and 180 min). The blood was collected by cardiac puncture, and the plasma was isolated by centrifugation. Then, the plasma was incubated with *P. aeruginosa* for an hour. The antibacterial activity was measured using a colony-forming unit assay. Data represent the means ± SD of more than three independent experiments.

**Figure 5 ijms-22-08243-f005:**
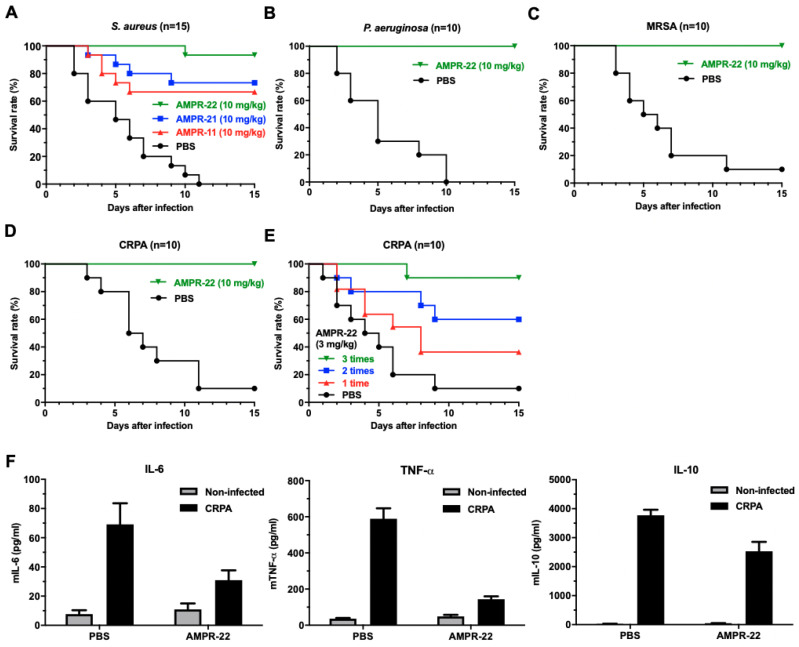
Antibacterial activity of AMPR-22 in a murine model of sepsis. (**A**) Survival rates of mice infected with *S. aureus*. Each group was treated with PBS, AMPR-11, or AMPR-11 analog peptides. *P. aeruginosa-* (**B**), MRSA- (**C**), or CRPA (**D**)-infected mice were treated with PBS or AMPR-22. All peptides were intravenously administered into mice at a single dose of 10 mg/kg. (**E**) The efficacy of multiple injections of the peptide. AMPR-22 (3 mg/kg) was injected into CRPA-infected mice one, two, or three times at 1 h intervals. Survival rates were calculated by observing the 10 mice in each group for 15 days (except the *S. aureus* (**A**) group, which had 15 mice). (**F**) Effects of AMPR-22 on host immune system in a murine model of sepsis. The cytokine levels of CRPA-infected mice were measured by ELISA. Plasma was collected 18 h after bacterial infection. PBS or AMPR-22 (10 mg/kg) was intravenously administered an hour after infection. Data represent the means ± SD of three independent experiments.

**Table 1 ijms-22-08243-t001:** MBC of the AMPR-11 and modified peptides.

Peptides	Sequence	MBC Value (µg/mL)			
		*S. aureus*	*P. aeruginosa*	MRSA	CRPA
AMPR-11 ^a^	KTMMQSGGTFGTFMAIGMGIR	100	100	100	110
***Substitution***					
K2	KKMMKSGGTFGTFMAIGMGIR	32	16	64	16
K3	KKMMKKGGTFGTFMAIGMGIR	16	8	32	8
K4	KKMMKKGGKFGTFMAIGMGIR	8	4	8	4
***Deletion***					
d58K	KMMKKGGKFGTFMAIGMGIR	8	2	2	8
d60M	KK MKKGGKFGTFMAIGMGIR	4	2	1	4
d62K	KKMM KGGKFGTFMAIGMGIR	8	2	8	8
d64G	KKMMKK GKFGTFMAIGMGIR	2	2	1	4
d66K	KKMMKKGG FGTFMAIGMGIR	8	4	4	4
d67F	KKMMKKGGK GTFMAIGMGIR	8	4	4	8
d68G	KKMMKKGGKF TFMAIGMGIR	4	4	4	8
d69T	KKMMKKGGKFG FMAIGMGIR	2	4	1	4
d70F	KKMMKKGGKFGT MAIGMGIR	256	128	128	>256
d71M	KKMMKKGGKFGTF AIGMGIR	4	2	4	8
d72A	KKMMKKGGKFGTFM IGMGIR	2	2	2	4
d73I	KKMMKKGGKFGTFMA GMGIR	256	>256	>256	>256
d74G	KKMMKKGGKFGTFMAI MGIR	4	4	4	8
d75M	KKMMKKGGKFGTFMAIG GIR	2	2	2	4
d76G	KKMMKKGGKFGTFMAIGM IR	4	4	4	4
d77I	KKMMKKGGKFGTFMAIGMG R	8	4	4	8
d78R	KKMMKKGGKFGTFMAIGMGI	16	4	4	16

^a^ Lee et al., 2020, *mBio*, 11(2): e03258-19. *S. aureus*, *Staphylococcus aureus*; *P. aeruginosa*, *Pseudomonas aeruginosa*; *MRSA*, *Methicillin-resistant S. aureus*; CRPA, Carbapenem-resistant *P. aeruginosa*.

**Table 2 ijms-22-08243-t002:** MBC of AMPR-21 and AMPR-22 against various bacterial species.

Group	Bacteria	Strains	MBC Value (µg/mL)		
			AMPR-11 ^a^	AMPR-21	AMPR-22
**Gram (+)**	*Staphylococcus aureus*	ATCC 29213	100	4	2
	*Bacillus subtilis*	ATCC 6633	90	2	1
	*Enterococcus faecium*	ATCC 19434	85	8	4
	*Streptomyces sindenensis*	ATCC 23963	95	4	2
	*Enterococcus faecalis*	ATCC 19433	85	8	4
	*Streptococcus pneumoniae*	NCCP 14585	100	8	4
**Gram (−)**	*Escherichia coli*	ATCC 25922	85	4	1
	*Pseudomonas aeruginosa*	ATCC 27853	100	4	2
	*Klebsiella pneumoniae*	ATCC 13883	100	4	1
	*Acinetobacter baumannii*	ATCC 19606	100	4	1
	*Enterobacter aerogenes*	ATCC 13048	90	4	2
**Multidrug** **Resistance**	Methicillin-resistant *S. aureus*	ATCC 33591	100	8	2
	Carbapenem-resistant *P. aeruginosa*	− ^b^	110	4	4
	Carbapenem-resistant *A. baumannii*	− ^b^	110	2	1
	Carbapenem-resistant *K. pneumoniae*	− ^b^	100	8	4
	Vancomycin-resistant *S. aureus*	NCCP 15872	120	8	4
	Vancomycin-resistant *E. faecium*	NCCP 11522	100	8	4

^a^ Lee et al., 2020, *mBio*, 11(2): e03258-19. ^b^ Clinically isolated at Korea University Hospital (Institutional Review Board approval no. 2015AN0129).

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
