# Peer review of "A Novel Peptide Derived from the Transmembrane Domain of Romo1 Is a Promising Candidate for Sepsis Treatment and Multidrug-Resistant Bacteria"

_ijms, 2021, doi:10.3390/ijms22158243_

Round 1

Reviewer 1 Report

Deok-gyun You and colleagues characterize the antimicrobial potential and the mode of aciton of Romo1-derived peptides. Among them, AMPR22 is the most promising, displaying efficacy in a mice model of sepsis.

Despite the significance of this final result, however, the paper can not be published as it is.

First of all, the quality of most of the figures in extremely poor and is not accetable for publication. In some cases the labels can be barely read. This is very annoying for the reader.

Second, few methodological points must be addressed.

MBC:

authors claim to calculate the minimum bactericidal concentration, however they misuse this term, as they perform simply a killing test, and not a proper conventional MBC assay.

Moreover, author test the efficacy of the peptides in in 10 mM sodium phosphate buffer 387 (pH 7.4) with a 1% volume of TS broth. Under these conditions bacteria are starving and are under osmotic stress. These conditions emphasise the effect of any antimicrobial compound, exposing the authors to the risk of artifacts, and, on the other hand, exposing fast readers to overestimate the antimicrobial effect of the compounds.

CD:

The signal may be hampered by the high absorbance of chloride ions, very abundand in PBS. Authors need to test the peptide in the presence only of sodium-phosphate buffer in order to claim any structuration in pbs. 

Moreover, there is no always corrispondence between the figure and the fig legend.

Spectrum and mode of action:

authors mix differnet techniques, there is no simmetry or coherence among the used concentrations and techniques. Moreover, the peptide concentrations selected for some of the experiments are far away from the concentrations displaying biological activity. This sections must be completely re-organized with more clarity, linearity and uniformity of conditions.

Cytotoxicity:

panel D: authors show results of the effect of 100 mg/kg stating that this amount is 10x the active one, however this is shown only in the following paragraph. This should be avoided.

Antimicrobial activity:

data are presented in a quite uncommon and compicated way. Authors may consider to present data in a different manner.

Author Response

July 21, 2021

Title: A novel peptide derived from the transmembrane domain of Romo1 is a promising candidate for sepsis treatment and multidrug-resistant bacteria

Manuscript ID: ijms-1280207

Thank you very much for your kind suggestions. We would like to resubmit our manuscript to International Journal of Molecular Sciences. The manuscript has been modified as the reviewers recommended, and revisions are shown in red.

We thank you for your continued interest in our work and for your consideration of our manuscript for publication in International Journal of Molecular Sciences.

Best regards,

Young Do Yoo, Ph. D.

Professor

Laboratory of Molecular Cell Biology, Graduate School of Medicine, Korea University College of Medicine, Korea University, 73, Goryeodae-ro, Seongbuk-gu, Seoul 02841, Republic of Korea.

Phone: 82-2-2286-1362

Responses to Reviewer Comments

Reviewer 1

[Comment 1]

the quality of most of the figures in extremely poor and is not acceptable for publication. In some cases the labels can be barely read. This is very annoying for the reader.

[Author response]

We apologize for these problems. We have improved the quality of all the figures. The labels of the figures and tables were also modified (line 104, 119, 168, 170, 197, 230, and 271).

[Comment 2] few methodological points must be addressed.

2-1. MBC: authors claim to calculate the minimum bactericidal concentration, however they misuse this term, as they perform simply a killing test, and not a proper conventional MBC assay. Moreover, author test the efficacy of the peptides in in 10 mM sodium phosphate buffer 387 (pH 7.4) with a 1% volume of TS broth. Under these conditions bacteria are starving and are under osmotic stress. These conditions emphasise the effect of any antimicrobial compound, exposing the authors to the risk of artifacts, and, on the other hand, exposing fast readers to overestimate the antimicrobial effect of the compounds.

[Author response]

In this study, we performed the assay of the minimum bactericidal concentration (MBC) with the physiological buffer instead of the bacterial broth media. The authors agree that sodium phosphate buffer (SPB) could be a poor condition for bacterial growth as the first reviewer pointed out. However, bacteria were well grown on the tryptic soy agar plates after 1 h incubation with SPB buffer as described in the method section. In this study, the control experiment (the bacterial growth after SPB incubation without the peptide) was also performed. Moreover, the assay in this manuscript is slightly modified from the protocols reported by many groups [1-4]. The authors revised some of sentences related to MBC more specifically to prevent the misunderstanding of the readers (line 94-95, 135, and 165).

2-2. CD: The signal may be hampered by the high absorbance of chloride ions, very abundand in PBS. Authors need to test the peptide in the presence only of sodium-phosphate buffer in order to claim any structuration in pbs. Moreover, there is no always corrispondence between the figure and the fig legend.

[Author response]

We agree with a reviewer’s comment regarding the possibility of hampered signal by chloride ions of PBS in circular dichroism (CD) spectroscopy. All the CD results in this study were presented after considering the possible artifacts by the buffer. Before the peptide analysis, we measured the signals of all kind buffers (without peptides) as described in the method section (line 396-398), and the raw data was corrected by re-calculation with an average buffer baseline spectra over the same wavelength regions. We added some explanations of CD spectroscopy into the figure legend (line 127-128) for the better understanding by the readers. And the authors apologize for the some uncorrelations between figure 1 and its legend. We corrected the legend of figure 1 (line 125-126).

2-3. Spectrum and mode of action: authors mix differnet techniques, there is no simmetry or coherence among the used concentrations and techniques. Moreover, the peptide concentrations selected for some of the experiments are far away from the concentrations displaying biological activity. This sections must be completely re-organized with more clarity, linearity and uniformity of conditions.

[Author response]

In the section of the mode of action study (figure 2), the concentrations of the peptide used were determined by considering the purpose of each experiment. Panels A and B are the comparative analysis of three kinds of peptides. Because the MBC value of AMPR-11 is higher than other peptides, we performed propidium iodine (PI) uptake assay with higher concentration of peptide. We used 20 µM of peptides in this assay even though it is slightly higher concentration compared to the biological assay. In the case of the electron microscopy (panel F and G), it was, unfortunately, needed high number of bacteria to be assayed. Therefore, as bacterial number was increased, we used the higher concentration of peptides, which was even far away from physiological conditions. However, this was within the acceptable ranges that are widely used [7-9].

2-4. Cytotoxicity: panel D: authors show results of the effect of 100 mg/kg stating that this amount is 10x the active one, however this is shown only in the following paragraph. This should be avoided.

[Author response]

The authors apologize for this lack of information about peptide concentration in Figure 3 panel D. As reviewer recommended, we have added the additional explanations into figure legend about concentration (line 205).

2-5. Antimicrobial activity: data are presented in a quite uncommon and compicated way. Authors may consider to present data in a different manner.

[Author response]

Revision Letter Figure 1. The schematic diagrams of antimicrobial activity assessments.

It has been reported that antimicrobial peptides have a short half-life in human body because of a protease attack or a renal clearance [10-13]. In this study, we investigated for the possibility of use as peptide antibiotics by estimating in vivo half-life of AMPR-22 with variety of approaches. However, we apologize for an insufficient explanation which was difficult to understand. Therefore, we have prepared additional explanations as follows. First, we investigated AMPR-22 activity in human or mouse blood component in vitro. The results from Figure 4A - 4D in the original manuscript were in vitro experiments for measurements of AMPR-22 activity affected by human and mouse blood factors. The antibacterial activity of AMPR-22 was gradually decreased in the presence of mouse plasma and human serum. But the time point to reach 50% inhibitory concentration (IC50) was extended compared to AMPR-11, which we previously reported [14]. In panel C and D, it was not affected by each of representative blood components. We have also presented the strategy of experiments for the assessments of AMPR-22 activity into Revision Letter Figure 1.

REFERENCES

  1. Jung, S.; Mysliwy, J.; Spudy, B. r.; Lorenzen, I.; Reiss, K.; Gelhaus, C.; Podschun, R.; Leippe, M.; Grötzinger, J., Human β-defensin 2 and β-defensin 3 chimeric peptides reveal the structural basis of the pathogen specificity of their parent molecules. Antimicrobial agents and chemotherapy 2011, 55, (3), 954-960.
  2. Maisetta, G.; Di Luca, M.; Esin, S.; Florio, W.; Brancatisano, F. L.; Bottai, D.; Campa, M.; Batoni, G., Evaluation of the inhibitory effects of human serum components on bactericidal activity of human beta defensin 3. Peptides 2008, 29, (1), 1-6.
  3. Nguyen, L. T.; de Boer, L.; Zaat, S. A.; Vogel, H. J., Investigating the cationic side chains of the antimicrobial peptide tritrpticin: hydrogen bonding properties govern its membrane-disruptive activities. Biochimica et Biophysica Acta (BBA)-Biomembranes 2011, 1808, (9), 2297-2303.
  4. Ouhara, K.; Komatsuzawa, H.; Kawai, T.; Nishi, H.; Fujiwara, T.; Fujiue, Y.; Kuwabara, M.; Sayama, K.; Hashimoto, K.; Sugai, M., Increased resistance to cationic antimicrobial peptide LL-37 in methicillin-resistant strains of Staphylococcus aureus. Journal of Antimicrobial Chemotherapy 2008, 61, (6), 1266-1269.
  5. Epand, R. F.; Pollard, J. E.; Wright, J. O.; Savage, P. B.; Epand, R. M., Depolarization, bacterial membrane composition, and the antimicrobial action of ceragenins. Antimicrobial agents and chemotherapy 2010, 54, (9), 3708-3713.
  6. Ma, L.; Ye, X.; Sun, P.; Xu, P.; Wang, L.; Liu, Z.; Huang, X.; Bai, Z.; Zhou, C., Antimicrobial and antibiofilm activity of the EeCentrocin 1 derived peptide EC1-17KV via membrane disruption. EBioMedicine 2020, 55, 102775.
  7. González-Pérez, C.; Tanori-Cordova, J.; Aispuro-Hernández, E.; Vargas-Arispuro, I.; Martínez-Téllez, M., Morphometric parameters of foodborne related-pathogens estimated by transmission electron microscopy and their relation to optical density and colony forming units. Journal of microbiological methods 2019, 165, 105691.
  8. Hartmann, M.; Berditsch, M.; Hawecker, J.; Ardakani, M. F.; Gerthsen, D.; Ulrich, A. S., Damage of the bacterial cell envelope by antimicrobial peptides gramicidin S and PGLa as revealed by transmission and scanning electron microscopy. Antimicrobial agents and chemotherapy 2010, 54, (8), 3132-3142.
  9. Kumar, P.; Kandi, S. K.; Manohar, S.; Mukhopadhyay, K.; Rawat, D. S., Monocarbonyl curcuminoids with improved stability as antibacterial agents against Staphylococcus aureus and their mechanistic studies. ACS Omega 2019, 4, (1), 675-687.
  10. Seo, M.-D.; Won, H.-S.; Kim, J.-H.; Mishig-Ochir, T.; Lee, B.-J., Antimicrobial peptides for therapeutic applications: a review. Molecules 2012, 17, (10), 12276-12286.
  11. Koo, H. B.; Seo, J., Antimicrobial peptides under clinical investigation. Peptide Science 2019, 111, (5), e24122.
  12. Fjell, C. D.; Hiss, J. A.; Hancock, R. E.; Schneider, G., Designing antimicrobial peptides: form follows function. Nature reviews Drug discovery 2012, 11, (1), 37-51.
  13. Deslouches, B.; Montelaro, R. C.; Urish, K. L.; Di, Y. P., Engineered cationic antimicrobial peptides (eCAPs) to combat multidrug-resistant bacteria. Pharmaceutics 2020, 12, (6), 501.
  14. Lee, H.-R.; You, D.-g.; Kim, H. K.; Sohn, J. W.; Kim, M. J.; Park, J. K.; Lee, G. Y.; Yoo, Y. D., Romo1-derived antimicrobial peptide is a new antimicrobial agent against multidrug-resistant bacteria in a murine model of sepsis. Mbio 2020, 11, (2), e03258-19.

Reviewer 2 Report

  1. Please write the Latin words in italics  (e.g. lines 41, 42, 44, 77 and so on).
  2. Line 94-95: there is: 'a minimum bacterial concentration' while it should be 'a minimum bactericidal concentration'.
  3. Figure 2. is interesting but illegible due to too small font size (especially part A)
  4. If you separate the G +, G- and multidrug bacteria groups in table 2, it will be clearer.
  5. Make Fig. 3, 4, 5 better quality. Again, it is interesting but difficult to read.
  6. My only serious concern to this manuscript is that you used peptides with, as you stated, purity of at least 75%. How do you know the antibacterial effect is caused by the peptide itself? Perhaps the other part of the sample (because maybe it is as much as 25%) is responsible for that?

Author Response

July 21, 2021

Title: A novel peptide derived from the transmembrane domain of Romo1 is a promising candidate for sepsis treatment and multidrug-resistant bacteria

Manuscript ID: ijms-1280207

Thank you very much for your kind suggestions. We would like to resubmit our manuscript to International Journal of Molecular Sciences. The manuscript has been modified as the reviewers recommended, and revisions are shown in red.

We thank you for your continued interest in our work and for your consideration of our manuscript for publication in International Journal of Molecular Sciences.

Best regards,

Young Do Yoo, Ph. D.

Professor

Laboratory of Molecular Cell Biology, Graduate School of Medicine, Korea University College of Medicine, Korea University, 73, Goryeodae-ro, Seongbuk-gu, Seoul 02841, Republic of Korea.

Phone: 82-2-2286-1362

Responses to Reviewer Comments

Reviewer 2

[Comment 1]

Please write the Latin words in italics (e.g. lines 41, 42, 44, 77 and so on).

[Author response]

We apologize for these errors. We corrected the Latin words into italics (mainly bacteria gene names) and marked in red (line 41-44, 62, 77, 82, 99, 147-148, 163, 192, 212, 219, 229, 252, 255, 288, 335-336, 339, 342, 366-368, 370-371, 472, 475, and 492).

[Comment 2]

Line 94-95: there is: 'a minimum bacterial concentration' while it should be 'a minimum bactericidal concentration'.

[Author response]

We apologize for this mistake. As the reviewer pointed out, we have corrected the word ‘bacterial’ into ‘bactericidal’ and marked in red (line 95).

[Comment 3]

Figure 2. is interesting but illegible due to too small font size (especially part A)

[Comment 4]

If you separate the G +, G- and multidrug bacteria groups in table 2, it will be clearer.

[Comment 5]

Make Fig. 3, 4, 5 better quality. Again, it is interesting but difficult to read.

[Author responses to comment 3-5]

We apologize again for these errors and the incomplete verification in the submitted manuscript. We have corrected all the figures with improved quality images and re-arranged. Additionally, the tables were modified (line 104, 120, 168, 171, 198, 231, and 272).

[Comment 6]

My only serious concern to this manuscript is that you used peptides with, as you stated, purity of at least 75%. How do you know the antibacterial effect is caused by the peptide itself? Perhaps the other part of the sample (because maybe it is as much as 25%) is responsible for that?

[Author response]

As the second reviewer pointed out, the peptides which used in this study were purified over the 75% purity by a high-performance liquid chromatography. These peptides were chemically synthesized using the Fmoc solid-phase synthesis method and the final solvent was distilled water. To confirm the antimicrobial activity of the peptide used in this study, we also purchased the peptide of 95% and we found that there was no significant difference in the antimicrobial activity between 75% and 95%.

Round 2

Reviewer 1 Report

The authors, although having generally ameliorated the paper, still did not address some important points that were previously indicated.

Maybe the major point is the CD experiments. These experiments have been badly performed and must be corrected or removed from the paper. The authors may have performed the blank with the buffer, however the signal of the peptide under 200 nm in the presence of chloride still remains not reliable.

Moreover, the spectra in PBS in my opinion are those of a random-coil peptide and not of a helix, which by the way is perfectly normal for alpha-helix peptides. With he exception of LL-37 that structures as helix also in acqueous solution, indeed most AMPs structure themself as helix only in the presence of membranes, SDS, TFE or liposomes. Authors may confront their spectra above 200 nm with those of other publication and will find confirmation of that.

Moreover, authors use a prediction software to calculate the % of helix, but the software may be misleading, or not properly used, so that their results are not logic. E.g.: a peculiar feature of alphahelix is a minimum around 220 nm. At this wavelength the value of the AMPR-22 is very different in PBS and in HFIP, however authors predict a similar amount of helix under both the conditions... and this is not logic. Moreover, the signal of minimum around 220 of the AMPR-21 in HFIP is lower than that of AMPR-21 in the same solvent, however authors predict more abundance of helix in the first than in the second, and this is not possible.

Author Response

July 26, 2021

Title: A novel peptide derived from the transmembrane domain of Romo1 is a promising candidate for sepsis treatment and multidrug-resistant bacteria

Manuscript ID: ijms-1280207

Dear Editor,

Thank you very much for your kind suggestions. We would like to resubmit our manuscript to International Journal of Molecular Sciences. The manuscript has been modified as the first reviewer recommended.

We thank you for your continued interest in our work and for your consideration of our manuscript for publication in International Journal of Molecular Sciences.

Best regards,

Young Do Yoo, Ph. D.

Professor

Laboratory of Molecular Cell Biology, Graduate School of Medicine, Korea University College of Medicine, Korea University, 73, Goryeodae-ro, Seongbuk-gu, Seoul 02841, Republic of Korea.

Phone: 82-2-2286-1362

Responses to Reviewer Comments

Reviewer 1

[Comment 1]

Maybe the major point is the CD experiments. These experiments have been badly performed and must be corrected or removed from the paper. The authors may have performed the blank with the buffer, however the signal of the peptide under 200 nm in the presence of chloride still remains not reliable.

[Author response]

In the present study, we performed circular dichroism (CD) spectroscopy with 50% Hexafluoro-2-propanol (HFIP) or phosphate buffered saline (PBS) to predict the secondary structure of the peptide. In the case of HFIP, it has been widely used for a peptide analysis as 2,2,2-Trifluoroethanol (TFE) or sodium dodecyl sulfate (SDS) [1, 2], and the predictions of a peptide secondary structure were performed in a web server (β-structure selection, BeStSel) [3]. However, CD data performed in PBS buffer might be confusing for the readers to understand the result as the first reviewer was indicated. Therefore, the CD results performed in PBS buffer were removed to prevent the misunderstanding of the readers as the first reviewer was suggested. We modified Figure 1 panel C and D and several sentences explaining CD data performed in PBS buffer were removed (line 115, 123, and 391).

REFERENCES

  1. Greenfield, N. J., Using circular dichroism spectra to estimate protein secondary structure. Nature protocols 2006, 1, (6), 2876-2890.
  2. Andersen, N. H.; Dyer, R. B.; Fesinmeyer, R. M.; Gai, F.; Liu, Z.; Neidigh, J. W.; Tong, H., Effect of hexafluoroisopropanol on the thermodynamics of peptide secondary structure formation. Journal of the American Chemical Society 1999, 121, (42), 9879-9880.
  3. Micsonai, A.; Wien, F.; Kernya, L.; Lee, Y.-H.; Goto, Y.; Réfrégiers, M.; Kardos, J., Accurate secondary structure prediction and fold recognition for circular dichroism spectroscopy. Proceedings of the National Academy of Sciences 2015, 112, (24), E3095-E3103.

Round 3

Reviewer 1 Report

.